# Quercetin and Quercitrin from *Agrimonia pilosa* Ledeb Inhibit the Migration and Invasion of Colon Cancer Cells through the JNK Signaling Pathway

**DOI:** 10.3390/ph15030364

**Published:** 2022-03-17

**Authors:** Nguyet-Tran Trinh, Thi Minh Ngoc Nguyen, Jong-In Yook, Sang-Gun Ahn, Soo-A Kim

**Affiliations:** 1Department of Biochemistry, Dongguk University College of Oriental Medicine, Gyeongju 38066, Korea; tranctu1994@gmail.com (N.-T.T.); ngocntm@dgu.ac.kr (T.M.N.N.); 2Department of Oral Pathology, Yonsei University College of Dentistry, Seoul 03722, Korea; jiyook@yuhs.ac; 3Department of Pathology, Chosun University College of Dentistry, Gwangju 61452, Korea; ahnsg@chosun.ac.kr

**Keywords:** migration, invasion, colon cancer, *Agrimonia pilosa* Ledeb, quercetin, quercitrin, JNK

## Abstract

Considering the high metastatic potential of colorectal cancer (CRC), the inhibition of metastasis is important for anti-CRC therapy. *Agrimonia pilosa* Ledeb (*A. pilosa*) is a perennial herbaceous plant that is widely distributed in Asia. The extracts of *A. pilosa* have shown diverse pharmacological properties, such as antimicrobial, anti-inflammatory, and antitumor activities. In the present study, the antimetastatic activity of *A. pilosa* was evaluated. Methanol extraction from the roots of *A. pilosa* was performed by high-performance liquid chromatography (HPLC) and 12 fractions were obtained. Among these, fraction 4 showed the most potent inhibitory effect on the migration of colon cancer cells. Using LC-HR MS analysis, quercetin and quercitrin were identified as flavonoids contained in fraction 4. Like fraction 4, quercetin and quercitrin effectively inhibited the migration and invasion of RKO cells. While the level of E-cadherin was increased, the levels of N-cadherin and vimentin were decreased by the same agents. Although they all activate the p38, JNK, and ERK signaling pathways, only SP600125, an inhibitor of the JNK pathway, specifically inhibited the effect of fraction 4, quercetin, and quercitrin on cell migration. An in vivo experiment also confirmed the antitumor activity of quercetin and quercitrin. Collectively, these results suggest that *A. pilosa* and its two flavonoids, quercetin and quercitrin, are candidates for the antimetastatic treatment of CRC.

## 1. Introduction

Colorectal cancer (CRC) is the second most deadly and the third most commonly diagnosed cancer worldwide, with more than 1.9 million new cases and 935,000 deaths recorded in 2020 [1]. Colorectal cancer develops slowly over several years, through the premalignant precursor adenoma to carcinoma sequence [2]. While the 5-year survival rate is higher than 90% in localized stage (stage I) patients, distantly spread tumor (stage IV) patients have a slightly higher than 10% 5-year survival rate, suggesting that metastasis is an important factor for CRC mortality [2,3]. Therefore, the early diagnosis and inhibition of metastasis are considered the most effective ways to inhibit tumor progression. 

Metastasis is a multistep process that includes the loss of cell-to-cell connection, invasion of the basal membrane and tissues, intravasation into the blood or lymphatic vessels, survival in the circulation system, and lastly, extravasation and proliferation at newly metastatic sites. Many studies have shown that the epithelial-mesenchymal transition (EMT) of epithelial malignancies plays an important role in the invasion and metastasis of tumor cells [4,5,6]. During EMT, epithelial cancer cells lose their polarity and adhesive phenotype and acquire mesenchymal characteristics with increased cell motility and invasiveness. Altered expression patterns of adhesion molecules are typical events in cancer cells. Switching from epithelial cadherin (E-cadherin), a cell-surface protein for epithelial junctions, to neural cadherin (N-cadherin) is a hallmark of EMT. Vimentin, an intermediate filament, is also highly expressed in mesenchymal cells and induces mesenchymal morphology and cell motility [7,8].

*Agrimonia pilosa* Ledeb (*A. pilosa*) is a herbaceous perennial plant that is widely distributed throughout Asia. It has been used to treat cancer, abdominal pain, bloody discharges, and eczema in traditional Chinese medicine [9,10]. Pharmacological studies have shown that extracts from *A. pilosa* have diverse pharmacological properties, such as antioxidant, anti-inflammatory, and antihyperglycemic activities [9,10,11]. Apigenin, catechin, hyperin, kaempferol, luteolin, quercetin, tiliroside, and tormentic acid are known constituents of *A. pilosa* [9]. Tiliroside, the major component of *A. pilosa* ethanol extract, showed anti-inflammatory activity by downregulating the expression levels of iNOS and COX-2 [12]. Several studies have also shown the antitumor activity of *A. pilosa* [13,14]. However, the active compound that enables the antitumor properties of *A. pilosa* and its mechanism of action are not yet understood. 

In this study, quercetin and quercitrin were identified in methanol extracts derived from the roots of *A. pilosa* using ultrahigh-performance liquid chromatography quadrupole time-of-flight high-resolution mass spectrometry (UPLC-Q-TOF-HRMS) analysis. The antitumor activities of fraction 4 from the extract of *A. pilosa*, quercetin, and quercitrin were assessed, and their molecular action mechanisms were also investigated in human colon cancer cells. By confirming their antitumor activity in xenograft nude mice, we suggest that quercetin and quercitrin should be explored as potent inhibitors of colon cancer metastasis. 

## 2. Results

### 2.1. Fraction 4 from A. pilosa Inhibits the Migration of Colon Cancer Cells

Components from the roots of *A. pilosa* were extracted with methanol and then separated into 12 fractions by HPLC. The anticancer activity of *A. pilosa* was evaluated by a wound healing assay. Among the derived fractions, fraction 4 strongly inhibited the migration of RKO colon cancer cells (Figure 1A). Compared with the untreated control and fraction 1, fraction 4 inhibited cell migration by 71% and 70%, respectively. Additionally, fraction 4 inhibited cell migration in a dose-dependent manner (Figure 1B). The effect of fraction 4 on cell migration was further confirmed in SW480 and HCT116 colon cancer cells. As shown in Figure 1C, fraction 4 also inhibited cell migration in both cell lines, demonstrating that the extract from *A. pilosa* effectively inhibits colon cancer cell migration.

### 2.2. Fraction 4 from A. pilosa Inhibited Cell Invasion and Colony Formation

Because fraction 4 showed a strong inhibitory effect on cell migration, we next examined the effect of fraction 4 on cell proliferation and cytotoxicity. During the 24-h incubation period, 50 μg/mL fraction 4 did not show a considerable effect on either cell proliferation or cytotoxicity, with 86.4 and 113.9%, respectively, compared with the untreated control (Figure 2A,B).

To further investigate the inhibitory effect of fraction 4 on cell migration, a transwell assay was performed. As shown in Figure 2C, fraction 4 significantly inhibited the number of invaded cells in a dose-dependent manner. In addition, fraction 4 largely inhibited colony formation, demonstrating its antitumor activity (Figure 2D). 

### 2.3. Identification of Quercetin and Quercitrin from Fraction 4 by UPLC-Q-TOF-HRMS

Because fraction 4 strongly inhibited cell migration, we next performed UPLC-Q-TOF-HRMS to identify the phytochemicals that affect cell migration in fraction 4. The phytochemicals catechin, luteolin, quercetin, apigenin, and kaempferol were reported to be major constituents of *A. pilosa* [9]. Using the known constituents of *A. pilosa* as a standard, quercetin and quercitrin were identified in fraction 4 (Figure 3A).

Interestingly, quercetin and quercitrin effectively inhibited RKO cell migration by 50 and 68%, respectively, at a concentration of 50 μM (Figure 3B). Furthermore, both quercetin and quercitrin largely inhibited cell invasion in a dose-dependent manner (Figure 3C). These data confirm quercetin and quercitrin as active compounds that have anticancer activity in fraction 4 from *A. pilosa*.

### 2.4. Quercetin and Quercitrin Induce Mesenchymal-to Epithelial Transition (MET)

EMT is a major process in the migration and invasion of cancer cells. From that point of view, MET is considered to be an important process for inhibiting the metastasis of cancer cells. To investigate whether quercetin and quercitrin play a role in MET, the expression levels of epithelial and mesenchymal markers were examined in RKO cells. As shown in Figure 4, the expression level of E-cadherin, a marker for epithelial cells, was increased by fraction 4, quercetin, and quercitrin in a dose-dependent manner. In contrast, the expression levels of vimentin and N-cadherin, markers for mesenchymal cells, were largely inhibited. These results suggest that fraction 4 from *A. pilosa* effectively induced the MET process and that this effect was achieved by quercetin and quercitrin. 

### 2.5. Quercetin and Quercitrin Inhibit Cell Migration through the JNK Signaling Pathway

To identify those signaling pathways involved in cell migration that are regulated by quercetin and quercitrin, we next investigated the effect of these flavonoids on the MAPK signaling pathways. The results of the Western blot analysis demonstrated that fraction 4 induced p38, JNK, and ERK phosphorylation in a dose-dependent manner (Figure 5A). As expected, both quercetin and quercitrin also induced the phosphorylation of p38, JNK, and ERK, similar to fraction 4. Fraction 4, quercetin, and quercitrin also activated MAPK signaling pathways in a time-dependent manner (data not shown). However, only SP600125 (an inhibitor of JNK) effectively restored the inhibitory effect of fraction 4, quercetin, and quercitrin, suggesting that quercetin and quercitrin play inhibitory roles in cell migration through the JNK signaling pathway (Figure 5B). Furthermore, when the cells were pretreated with SP600125, the induced level of E-cadherin by quercetin and quercitrin was completely restored to the control level. Additionally, SP600125 also restored the levels of N-cadherin and vimentin that were inhibited by quercetin and quercitrin to the control levels, suggesting that the induction of the MET by quercetin and quercitrin is mediated by the JNK signaling pathway (Figure 5C). 

### 2.6. Quercetin and Quercitrin Inhibit the Growth of Tumors In Vivo

To examine the anticancer activity of quercetin and quercitrin in vivo, 20 Balb/c nude mice were injected with RKO cells in the right flank, as described in Section 4: Materials and Methods. Seven days after cell injection, fraction 4, quercetin, and quercitrin were intraperitoneally injected every other day for a total of three times (*n* = 5 for each group). As shown in Figure 6B, body weight did not show differences between the untreated control and agent-treated groups. However, all of the tested agents effectively inhibited the growth of solid tumors in Balb/c nude mice (Figure 6C). Immunohistochemistry showed that quercetin and quercitrin treatment increased E-cadherin positive cells while decreasing vimentin positive cells, compared with the vehicle-treated control (Figure 6D). These results demonstrated the antitumor activity of quercetin and quercitrin in vivo.

## 3. Discussion

Several studies have reported the anticancer activity of extracts from *A. pilosa* through the induction of apoptotic cell death in hepatoma and osteosarcoma cells [14,15]. More recently, Eom and Kim reported that a methanol extract from *A. pilosa* inhibits cell invasion by decreasing the levels and activities of MMP-2 and MMP-9 in HT1080 human sarcoma cells [16]. Although the active compound was not identified, these reports suggest that *A. pilosa* contains a natural compound that has anticancer activity. In the present study, we demonstrated that the methanol extract of roots from *A. pilosa* strongly inhibited the migration of colon cancer cells. 

Although significant progress has been achieved in the diagnosis and treatment of colorectal cancer (CRC), there is no effective therapy for metastatic CRC. Metastasis is the main reason for the high mortality and morbidity observed in CRC patients, with more than 50% of patients developing liver metastases during their lifespan [2,5]. EMT is a critical process for the dissociation of cancer cells from primary carcinoma and the subsequent migration and dissemination to distant sites. Conversely, MET is associated with the loss of migration properties, with cells expressing junctional complexes, the hallmarks of epithelial cells. Therefore, MET is believed to be an important target for interrupting seeding metastasis because it inhibits migration [8,17,18]. Recently, phytochemicals with anti-EMT properties have attracted researchers’ attention because of their general low toxicity and low side effects. Several phytochemicals were identified as inhibitors of metastasis, such as fisetin, epigallocatechin gallate (EGCG), quercetin, resveratrol, and curcumin [19,20,21,22].

Quercetin (3,3′,4′,5,7-pentahydroxyflavone) is a well-known flavonoid contained in a variety of vegetables and fruits and that demonstrates anticancer activities, such as proapoptotic, antiproliferative, and antioxidant effects, in many cancer cells [23]. Previous studies have shown that quercetin has antimetastatic activity in several cancer cell types [24,25]. However, the effect of quercetin on the migration and invasion of colon cancer is still unclear. Quercitrin (3′,4′,5,7-tetrahydroxy-3-(α-L-rhamnopyranosyloxy)flavone) is a natural derivative of quercetin obtained by the substitution of an α-L-rhamnopyranosyl moiety at position 3 via a glycosidic bond (Figure 3A). Although quercitrin is known to have antioxidative, anti-inflammatory, and anticancer activities, to our knowledge there have been no reports related to its migratory activity [26,27]. Using UPLC-Q-TOF-HRMS, we identified quercetin and quercitrin as the active flavonoids, with strong antimetastatic activity in methanol extract from *A. pilosa* (Figure 3). 

Because quercetin has a lipophilic nature, it can cross the cell membrane and activate various cellular signaling pathways. Previous studies have shown that quercetin inhibits cell migration and invasion in various cancer cell types, including breast, osteosarcoma, hepatoma, and oral cancer cells, by inhibiting the expression levels and activities of MMP-2 and MMP-9 [24,25,28,29]. Quercetin has also been shown to regulate MMPs through the AP-1 or NF-κB signaling pathway, depending on the cell type [29,30,31]. This study is the first to show that not only quercetin but also quercitrin effectively inhibited colon cancer cell migration and invasion by inducing MET through the JNK signaling pathway (Figure 4 and Figure 5). 

In conclusion, we demonstrated that the methanol extract of roots from *A. pilosa* strongly inhibits colon cancer cell migration, and we identified quercetin and quercitrin as active phytochemicals with antimetastatic activity. Both quercetin and quercitrin effectively inhibited cell migration and invasion by inducing the MET through the JNK signaling pathway. An in vivo study also confirmed the anticancer activity of these compounds. Taken together, these findings suggest quercetin and quercitrin as potential chemopreventive drugs in therapeutic strategies for reducing colorectal cancer mortality.

## 4. Materials and Methods

### 4.1. Materials

Quercetin, SP600125, and SB202190 were purchased from Millipore Sigma Aldrich (St. Louis, MO, USA). Quercitrin was obtained from Supelco (Bellefonte, PA, USA) and U0126 was purchased from Cell Signaling Technology (Danvers, MA, USA). 

### 4.2. Cell Culture and Wound Healing Assay

RKO, SW480, and HCT116 human colon cancer cells were obtained from the American Type Culture Collection (ATCC, Manassas, VA, USA) and maintained in Dulbecco’s Modified Eagle’s medium (DMEM, Hyclone, UT, USA) supplemented with 10% FCS, 100 units/mL penicillin, and 100 μg/mL streptomycin (Gibco BRL, Grand Island, NY, USA) at 37 °C under a 5% CO_2_ atmosphere. Cells were seeded on 6-well plates at a density of 1.5 × 10^6^ cells/mL and incubated until the population reached confluence. The cell monolayer was scratched using a blue pipette tip and then washed with PBS twice. Fresh media was added, and the cells were treated with fractions from *A. pilosa*. Cell migration was observed at 0, 24, 48, and 72 h under a conventional microscope. 

### 4.3. Cell Proliferation and Cytotoxicity Assay

Cells were seeded on 12-well plates at a density of 1 × 10^6^ cells/mL. The cells were cultured overnight and then treated with various concentrations of fraction 4 from *A. pilosa*, quercetin, or quercitrin for 24 h. Cell viability was measured using a 3-(4,5-dimethylthiazol-2-yl)-2,5-diphenyltetrazolium bromide (MTT) assay according to a previously described method [32].

For the cytotoxicity assay, cells were seeded on 6-well plates at a density of 1 × 10^6^ cells/mL. The cells were cultured overnight and then treated with fraction 4 from *A. pilosa*, quercetin or quercitrin for 24 h. Cytotoxicity was evaluated by measuring lactate dehydrogenase (LDH) activity, using an LDH-Glo^TM^ cytotoxicity assay kit (Promega, Madison, WI, USA) according to the manufacturer’s instructions.

### 4.4. Cell Invasion Assay

Transwell invasion assays were performed using the Corning Transwell^®^ system (Merck KGaA, Darmstadt, Germany). RKO cells were cultured in serum-free media for 24 h and then seeded at a density of 1.5 × 10^6^ cells/mL on a 0.5% gelatin-coated upper chamber. Ten-percent FCS-containing cell culture supernatant was added to the lower chamber and then treated with fraction 4 from *A. pilosa*, quercetin, or quercitrin. After incubation for 72 h, the cells remaining on the upper side of the filter were removed with a cotton swab, and the cells that invaded through the membrane were fixed with 70% ethanol for 10 min. Finally, the cells were stained with 0.2% crystal violet and then counted under a microscope (Lionheart FX, BioTek, Winooski, VT, USA).

### 4.5. Colony Formation Assay

Cells were seeded on 6-well plates at a density of 5 × 10^3^ cells/mL. After 24 h, the cells were treated with fraction 4 from *A. pilosa*, quercetin, or quercitrin and then further incubated for 6 days until visible colonies were formed. Following fixation with methanol for 20 min, the colonies were stained with crystal violet for 5 min. The number of colonies was photographed with a camera and counted.

### 4.6. Plant Material, Extraction, Fractionation, and Identification of Quercetin and Quercitrin

*A. pilosa* was collected from Hongcheon-gun, Gangwon-do, Korea and identified by one of us (J.-I. Yook). The dried roots of *A. pilosa* were extracted with 100% methanol in a 50 °C shaking water bath for 3 days. The filtrated extracts were evaporated on a rotary evaporator and freeze-dried to obtain the crude extract. The extract was dissolved in methanol and then fractionated by high-performance liquid chromatography (HPLC) using an XTerra Prep MS C18 OBD column (KRICT, Daejeon, Korea). Fraction 4 from HPLC was further analyzed by UPLC-Q-TOF-HRMS, which was carried out on a SYNAPT G2-Si (Waters Corp., Milford, MA, USA) (KIST, Seoul, Korea). The data were qualitatively analyzed using MassLynx V4.1 software from Waters Corporation (Milford, MA, USA). Quercetin and quercitrin were used as standards.

### 4.7. Western Blot Analysis

RKO cells were treated with various concentrations of fraction 4 from *A. pilosa*, quercetin, or quercitrin for the indicated time periods. The cells were washed twice with cold PBS and lysed in RIPA buffer (PBS supplemented with 1% NP-40, 1 mM phenylmethylsulfonyl fluoride, 1 μg/mL aprotinin, and 1 mM sodium orthovanadate). The cell lysates were incubated at 4 °C for 30 min and then cleared by centrifugation at 10,000× *g* for 10 min. The total protein concentration was measured using Protein Assay Dye Reagent Concentrate (Bio-Rad Laboratories, Inc., Hercules, CA, USA). Protein samples (30 μg) were resolved by 6 or 10% SDS-PAGE, transferred onto nitrocellulose membranes, and then blocked with 5% skim milk at room temperature for 4 h. Immunoblotting was performed at 4 °C overnight using the following primary antibodies: anti-E-cadherin, anti-N-cadherin, anti-Vimentin, anti-ERK, anti-p-ERK, anti-p38, anti-p-p38, anti-JNK, anti-p-JNK (Cell Signaling Technology), and anti-β-actin (Sigma Aldrich). The blots were then incubated with secondary antibodies at room temperature for 1 h and detected using Immobilon^®^ Western Chemiluminescent HRP Substrate (Millipore Corporation, Billerica, MA, USA) or SuperSignal^TM^ West Femto Maximum Sensitivity Substrate (Thermo Scientific, Rockford, IL, USA).

### 4.8. Nude Mouse Xenograft Assay

Three-week-old male Balb/c nude mice were purchased from NaRa Biotech (Seoul, Korea). The mice were housed at a temperature of 25 °C with 50~60% relative humidity and were fed standard laboratory chow. After 1 week of adaptation, RKO cells (1.5 × 10^7^ cells/mL, 100 μL/mouse) were subcutaneously injected into the right flank midline. The volume of the tumors was measured every 2 days using a caliper, as follows: tumor volume = (length) × (width)^2^ × 0.5. When the length was ≥2 mm, the mice were randomly divided into 4 treatment groups, with 5 mice in each group: vehicle, fraction 4 from *A. pilosa*, quercetin, and quercitrin. Fraction 4, quercetin, and quercitrin were injected intraperitoneally at concentrations of 0.5 mg/10 g, 0.151 mg/10 g, and 0.224 mg/10 g, respectively. The vehicle group was injected with 0.1% DMSO in saline. Injections were performed every other day for a total of three times.

### 4.9. Immunohistochemistry

The excised tumors were fixed in 10% buffered formalin and embedded in paraffin. Tissue sections (4 μm thick) were deparaffinized in xylene and rehydrated in a series of graded ethanol solutions, then exposed to microwave radiation in a citrate buffer (pH 6.0) for 15 min. Endogenous peroxidase activity was blocked with 3% H_2_O_2_ in methanol for 5 min. The sections were washed three times with PBS and then incubated with goat serum for 20 min at room temperature. Primary antibodies (1:100 dilution) were incubated overnight at 4 °C. The sections were washed three times with PBS and then incubated with biotinylated secondary antibodies for 30 min at room temperature. Immune complexes were detected with streptavidin-peroxidase complex (Agilent, Santa Clara, CA, USA) and 3,3′-diaminobenzidine (Agilent). The sections were counterstained with hematoxylin, dehydrated in a graded series of alcohol solutions, and mounted in malinol (Muto Pure Chemicals, Tokyo, Japan).

### 4.10. Statistical Analysis

The data are expressed as the means ± SD. Differences between groups were analyzed by Student’s *t*-test and an ANOVA (GraphPad Prism, San Diego, CA, USA). The data were further analyzed using Dunnett’s post hoc test. *p* < 0.01 was considered statistically significant.

## Figures and Tables

**Figure 1 pharmaceuticals-15-00364-f001:**
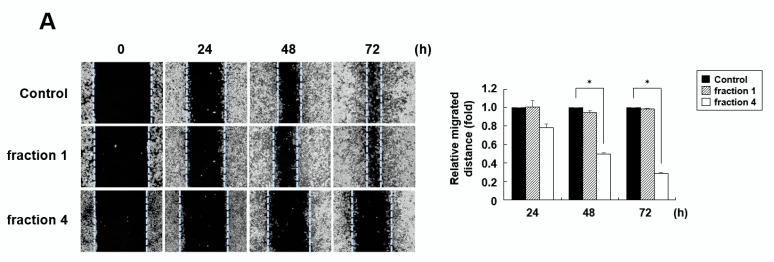
Fraction 4 from *A. pilosa* inhibits the migration of colon cancer cells. (**A**) RKO cells were treated with 50 μg/mL fraction 1 or 4 for 24, 48, or 72 h. (**B**) RKO cells were treated with the indicated concentrations of fraction 4 for 72 h. (**C**) SW480 and HCT116 cells were treated with 50 μg/mL of fraction 4 for 72 h. Wound healing was quantified by measuring the mean line of the wound edges. All experiments were performed at least three times. The data are presented as the means ± SD. * *p* < 0.01.

**Figure 2 pharmaceuticals-15-00364-f002:**
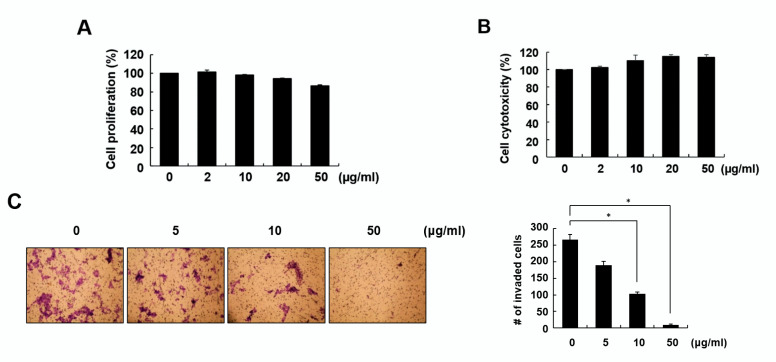
Effects of fraction 4 from *A. pilosa* on cell invasion and colony formation. (**A**,**B**) RKO cells were treated with the indicated concentrations of fraction 4 for 24 h. Cell proliferation was measured by MTT assay (**A**). Cell cytotoxicity was assessed by LDH assay (**B**). (**C**) Transwell assays were performed to measure the invasion of RKO cells. The cells were treated with the indicated concentrations of fraction 4 for 72 h. (**D**) RKO cells were treated with the indicated concentrations of fraction 4 for 6 days. Colonies were fixed and then stained with crystal violet. All experiments were performed at least three times and presented as the means ± SD. * *p* < 0.01.

**Figure 3 pharmaceuticals-15-00364-f003:**
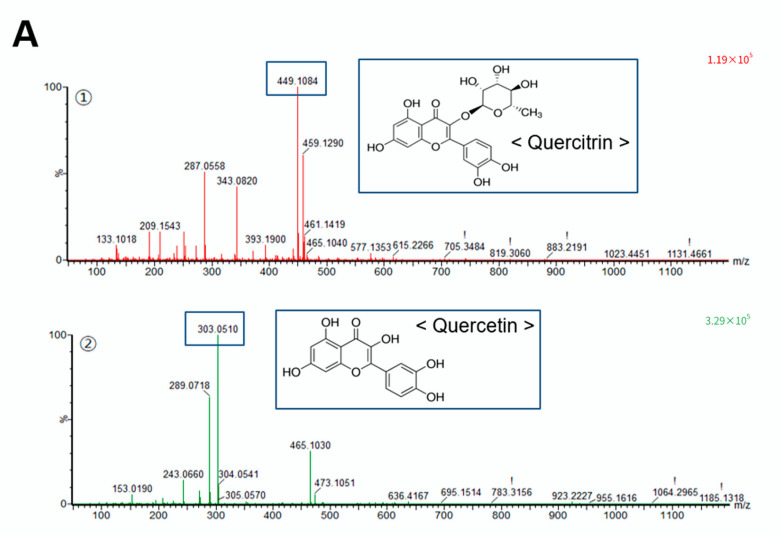
Identification of quercetin and quercitrin as active compounds in cell migration. (**A**) Fraction 4 from *A. pilosa* was further separated by UPLC, and then the mass spectrometer was operated in positive ion mode. RKO cells were treated with quercetin or quercitrin and the effects on cell migration were assessed by wound healing assays (**B**); transwell assays (**C**). * *p* < 0.01.

**Figure 4 pharmaceuticals-15-00364-f004:**
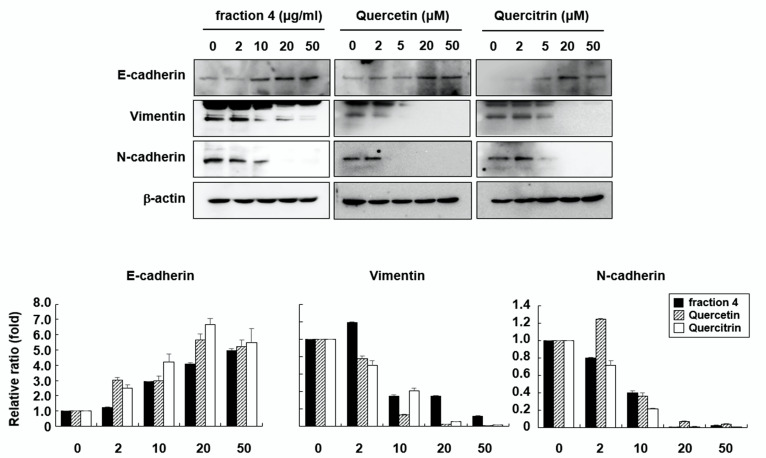
Fraction 4 from *A. pilosa* and its active compounds, quercetin and quercitrin, induce MET. RKO cells were treated with fraction 4, quercetin, or quercitrin for 24 h. Total cell lysates were prepared and the expression levels of E-cadherin, vimentin and N-cadherin were detected by Western blot analysis.

**Figure 5 pharmaceuticals-15-00364-f005:**
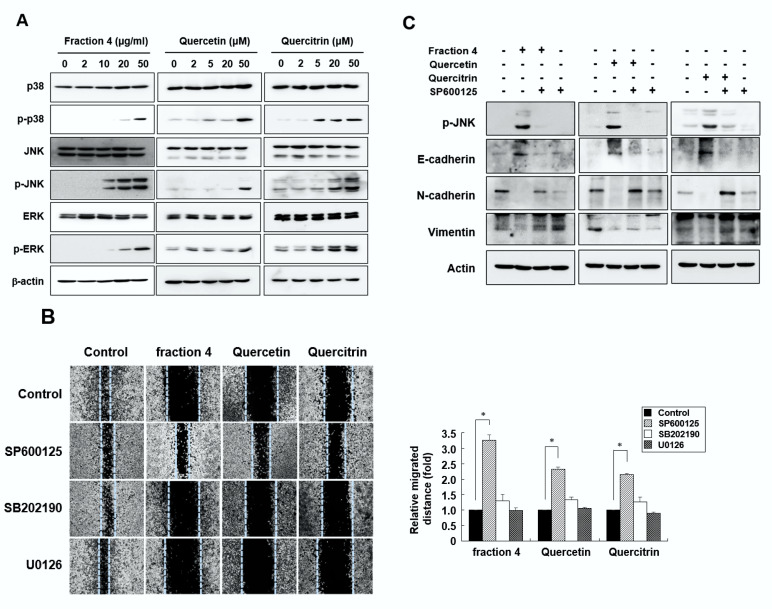
Quercetin and quercitrin inhibit cell migration through the JNK signaling pathway. (**A**) RKO cells were treated with fraction 4, quercetin, or quercitrin for 24 h. Total cell lysates were prepared and the levels of p38, p-p38, JNK, p-JNK, ERK, and p-ERK were detected by Western blot analysis. (**B**) The cells were pretreated with SP600125, SB202190, or U0126 for 2 h and then treated with fraction 4, quercetin, or quercitrin. After 72 h, wound healing was quantified by measuring the mean line of the wound edges. All experiments were performed at least three times. The data are presented as the means ± SD. (**C**) The cells were pretreated with SP600125 for 2 h and then with fraction 4, quercetin, or quercitrin for an additional 24 h. Total cell extracts were prepared and subjected to Western blot analysis. * *p* < 0.01.

**Figure 6 pharmaceuticals-15-00364-f006:**
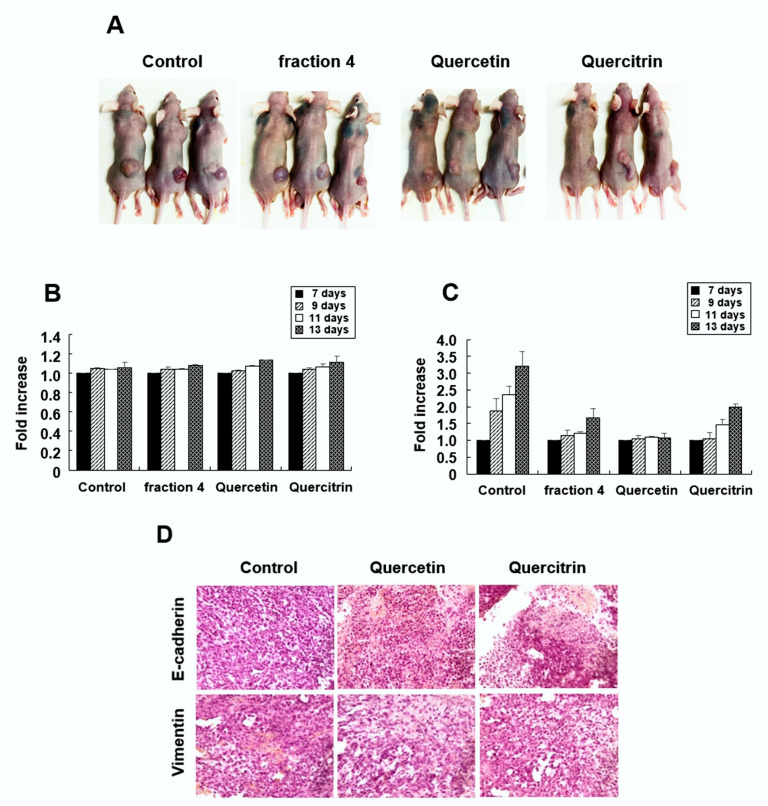
Antitumor effect of quercetin and quercitrin in vivo. RKO cells were subcutaneously injected into the right flanks of Balb/c nude mice. Fraction 4, quercetin, or quercitrin was injected intraperitoneally every other day, beginning on day 7. (**A**) Tumor images were obtained on day 13. (**B**) Serial body weights were measured from day 7 to day 13. (**C**) Serial tumor volumes were measured via caliper and calculated using the formula V = (ab^2^)/2, in which “a” is the largest diameter and “b” is the shortest diameter of the tumor. (**D**) Mice were sacrificed on day 13, and immunohistochemical staining for E-cadherin and vimentin was obtained from paraffin sections of solid tumor.

## Data Availability

Data is contained within the article.

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
