# Peer review of "Quercetin and Quercitrin from *Agrimonia pilosa* Ledeb Inhibit the Migration and Invasion of Colon Cancer Cells through the JNK Signaling Pathway"

_pharmaceuticals, 2022, doi:10.3390/ph15030364_

Round 1

Reviewer 1 Report

The comments:

The Authors did not include the consent of the bioethics committee

The Authors should compare the results using the normal cell line

The Authors should use a different statistical test

Author Response

Dear reviewer,

We sincerely appreciate your thoughtful comments and suggestions for the improvement of our manuscript ID: pharmaceuticals-1610707.

Please find following response to reviewer’ comments.

Title: Quercetin and quercitrin from Agrimonia pilosa Ledeb inhibit the migration and invasion of colon cancer cells through the JNK signaling pathway

  1. The Authors did not include the consent of the bioethics committee.

We got the consent of Animal Care and Use Committee (CIACUC 2020-A0014) and mentioned on page 12, line 344~346.

  1. The Authors should compare the results using the normal cell line.

In this study, we demonstrated quercetin an quercitrin which were identified in extracts from A. pilosa strongly inhibits colon cancer cell migration and invasion compared with untreated control cells. We believe the data clearly shows the fraction 4, quercetin and quercitrin effectively inhibit the colon cancer cell migration (Fig. 1A, 1B and 1C, Fig. 2C, Fig 3B, and 3C). Because normal cells are not migrated well itself, we cannot show the inhibitory effect of quercetin and quercitrin using normal cells.  

  1. The Authors should use a different statistical test.

Thanks for thoughtful comment. Data were analyzed by Student’s t-test and ANOVA. Additionally, we further analyzed using Dunnett’s post-hoc test. We mentioned this on ‘Materials and Methods’ section on page 11, line 335~336. By applying different statistical tests, the graph of Fig. 1A on page 2 was changed.

Additionally, we sent our manuscript to AJE, a professional language editing company, and edited following their instructions.

Thank you very much for reviewing our manuscript.

Sincerely yours,

Soo-A Kim, Ph.D.

Reviewer 2 Report

The manuscript has a topic of interest - colorectal cancer and its metastasis. The use of natural compounds with anti-carcinogenic and, in particular, anti-metastatic effects is welcome. Research in this field is an area of interest for the medical world. Agrimonia is a genus of about 15 species - rich in active compounds with diverse pharmacological effects.

The manuscript is written clear, relevant for the field and presented in a well-structured manner.

The cited references are correct - mostly within the last 5 years, and do not include an abnormal number of self-citations.

The experimental design is appropriate to test the hypothesis and the results are reproducible. After describing the in vitro experiments, the authors also described the in vivo experiments - by which they confirmed the previously obtained data.

The figures/images are appropriate and show the correct data.

The conclusions are clear and reasoned.

The authors respect the ethics and procedural conditions with animals. The authors declare the availability of the data - these are available from the corresponding author.

Author Response

Dear reviewer,

We sincerely appreciate your thoughtful comments for our manuscript ID: pharmaceuticals-1610707.

We sent our manuscript to AJE, a professional language editing company, and edited following their instructions.

Thank you very much for reviewing our manuscript.

Sincerely yours,

Soo-A Kim, Ph.D.

Reviewer 3 Report

In this manuscript, Trinh et al. described a study, in which they tried to determine whether Agrimonia pilosa Ledeb is a potent inhibitor of cell migration and invasion in colorectal cancer cells. They found that this herb extract and its components quercetin and quercitrin could not only inhibit cell migration and invasion in vitro but repress tumor growth in mice models. The results in this study are quite convincing and provide a novel insight for the application of Agrimonia pilosa Ledeb. Two specific comments are listed below.

  1. The in vitro results indicate that the extracts of Agrimonia pilosa Ledeb have inhibitory effects on cell migration and invasion but not on cell proliferation, which means they have no cytotoxic effect on colon cancer cells. How do the authors explain that the mouse results showed a significant anti-tumorigenic effect for treatments of those compounds?
  2. Furthermore, based on the in vitro results of this study, the authors should perform the tail vein metastatic assay, but not xenograft tumor growth assay in the mouse model.

Author Response

Dear reviewer,

We sincerely appreciate your thoughtful comments and suggestions for the improvement of our manuscript ID: pharmaceuticals-1610707.

Please find following response to reviewer’ comments.

Title: Quercetin and quercitrin from Agrimonia pilosa Ledeb inhibit the migration and invasion of colon cancer cells through the JNK signaling pathway

  1. The in vitro results indicate that the extracts of Agrimonia pilosaLedeb have inhibitory effects on cell migration and invasion but not on cell proliferation, which means they have no cytotoxic effect on colon cancer cells. How do the authors explain that the mouse results showed a significant anti-tumorigenic effect for treatments of those compounds?

Thank you for your thoughtful comment. We also deeply considered this point.

When the cells were treated with 50 μg/ml of fraction 4 from A. pilosa (a concentration used for migration assay) for 24 h, cell proliferation was 86.4% compared with untreated control (Fig. 2A). Because, all the wound healing assays and transwell assays were performed for 3 days, we further examined the cell proliferation assay for 3 days and shown below (Fig. Aa). Although cell proliferation was gradually decreased depending on the incubation period, quercetin or quercitrin treated cells showed cell proliferation 75.4 and 79.9% on 72 h treatment compared with untreated control cells.

However, colony formation assay which treated agents for 6 days showed strong anti-tumor activity at the concentration of 50 μg/ml for fraction 4 and 50 μM of quercetin or quercitrin, suggesting these agents have anti-tumor activity for long period treatment (Fig. 2D and Fig. Ab). Furthermore, 25 μg/ml of fraction 4 and 25 μM of quercetin or quercitrin didn’t show anti-tumor activity on colony formation assay (Fig. 2D and Fig. Ab), but showed strong inhibitory effect on cell migration (Fig. 1B). These data clearly demonstrate the inhibitory effect of quercetin and quercitrin on cell migration.

  1. Furthermore, based on the in vitro results of this study, the authors should perform the tail vein metastatic assay, but not xenograft tumor growth assay in the mouse model.

As reviewer mentioned, tail vein metastatic assay is better method for this study. However, we choose Balb/C nude mice for xenograft tumor growth assay because we are planning to examine the anti-tumor activity of agents for further study. At the end of in vivo assay, we excised the tumors and performed immunohistochemistry for E-cadherin and vimentin. Results showed quercetin and quercitrin treatment increased E-cadherin positive cells while decreased vimentin positive cells compared with vehicle-treated control. We added Immunohistochemistry data in ‘Result’ section on page 7, line 164~168, ‘Fig. 6D’ on page 8, ‘Figure legend’ on page 9, line 195~196, and ‘Materials and methods’ section on page 11, line 320~332.

In this study, we want to focus the anti-metastatic activity of quercetin and quercitrin and we are planning to study anti-cancer activity of these agents for further study. Please understand our situation.

Additionally, we sent our manuscript to AJE, a professional language editing company, and edited following their instructions.

Thank you very much for reviewing our manuscript.

Sincerely yours,

Soo-A Kim, Ph.D.

Round 2

Reviewer 3 Report

All the questions have been answered adequately. This manuscript is nearly suitable for publication.